# Practice-Based Research in Complementary Medicine: Could N-of-1 Trials Become the New Gold Standard?

**DOI:** 10.3390/healthcare8010015

**Published:** 2020-01-08

**Authors:** Joanne Bradbury, Cathy Avila, Sandra Grace

**Affiliations:** 1School of Health and Human Sciences, Southern Cross University, Gold Coast, QLD 4225, Australia; 2School of Health and Human Sciences, Southern Cross University, Lismore, NSW 2480, Australia; cathy.avila@scu.edu.au (C.A.); sandra.grace@scu.edu.au (S.G.)

**Keywords:** N-of-1 trials, complementary medicine, levels of evidence, practice-based research, naturopathic medicine

## Abstract

Complementary medicines and therapies are popular forms of healthcare with a long history of traditional use. Yet, despite increasing consumer demand, there is an ongoing exclusion of complementary medicines from mainstream healthcare systems. A lack of evidence is often cited as justification. Until recently, high-quality evidence of treatment efficacy was defined as findings from well-conducted systematic reviews and meta-analyses of randomized controlled trials. In a recent and welcome move by the Oxford Centre for Evidence-Based Practice, however, the N-of-1 trial design has also been elevated to the highest level of evidence for treatment efficacy of an individual, placing this research design on par with the meta-analysis. N-of-1 trial designs are experimental research methods that can be implemented in clinical practice. They incorporate much of the rigor of group clinical trials, but are designed for individual patients. Individualizing treatment interventions and outcomes in research designs is consistent with the movement towards patient-centered care and aligns well with the principles of holism as practiced by naturopaths and many other complementary medicine practitioners. This paper explores whether rigorously designed and conducted N-of-1 trials could become a new ‘gold standard’ for demonstrating treatment efficacy for complementary medicine interventions in individual patients in clinical practice.

## 1. Introduction

A recent overview of systematic reviews reported that there was a lack of evidence for a number of complementary medicine therapies in any health condition [1]. This overview only included evidence from systematic reviews. In reality, the majority of complementary medicine interventions have not been evaluated using randomized controlled trial designs and would therefore not be included in a systematic review. Historically, knowledge about complementary medicine interventions, particularly in the Western herbal medicine tradition, was handed down through an oral tradition based on direct observation by practitioners over long periods of time.

Nevertheless, the general statement that there is no evidence of an effect implies that there is evidence of no effect. The two statements have very different meanings. The overview itself acknowledges that there was a lack of randomized controlled trials in complementary medicine. Another major issue cited was the large number of small, underpowered ‘pilot’ studies in complementary medicine that were statistically inconclusive. The report also acknowledged the difficulty in masking the intervention in complementary medicine, thus, blinding was an issue that contributed to increasing the risk of bias and therefore downgrading the quality of evidence. Yet another issue was that many studies involved more than one treatment, making it difficult to determine which treatment caused the effect. Indeed, in practice, many complementary medicine practitioners acknowledge the value of the ‘therapeutic relationship’ and the difficultly of separating it -from a treatment intervention, as would be required by in a reductive, rigorously-conducted randomized controlled trial.

Personalizing treatment interventions and outcomes for each patient is an approach consistent with the holistic approach that is used by many health practitioners, particularly those from complementary medicine professions (such as naturopathy, Western herbal medicine, osteopathy, acupuncture, and homeopathy). Two patients with sleeping problems presenting to a naturopath, for example, could receive two very different treatment regimens, depending on the particular patient history; it is the whole person who is placed at the center of the treatment regime, in addition to the presenting symptoms and/or disease diagnosis. In naturopathic philosophy this is known as ‘Treat the Whole Person’ (*tolle totum*) [2]. Indeed, this principle of holism has historically underpinned and united all the complementary medicine professions [3].

Holistic practitioners are interested in the functioning of the person within their broader social, economic and cultural context, including their inner psycho-social-spiritual wellbeing. As such, the focus of the treatment consultation goes beyond the presenting symptom(s) and its medical history. This holistic approach is an important guiding philosophy that unites the various complementary medicine professions under the umbrella of complementary medicine and at the same time distinguishes complementary medicine from conventional medicine. In terms of producing evidence of efficacy, however, this holistic approach is problematic as it is, by definition, not reductionist, and therefore, finds itself at ideological odds with the aims of the gold-standard clinical trials methodology—to reduce the treatment effect to a single, measurable parameter.

Randomized controlled trials (RCT) are highly controlled experimental designs that generally aim to include homogenous clinical samples to test for specific effects of a single intervention. Heterogeneity in the sample increases variance and standard error sizes, which reduces the probability of finding a significant result. A limitation of RCT designs is that the sometimes stunning effects demonstrated in such tightly controlled clinical trials cannot be replicated in clinical practice where heterogeneity exists both within (e.g., comorbidities and polypharmacy) and between patients [4,5].

Pragmatic trial designs that allow for heterogeneity in the sample must be incorporated into the future of clinical trial designs to facilitate the translation of evidence into practice. In an era where treatments are shifting towards personalized and precision medicine, there is a real need for the production of high-quality evidence about individual patients to guide clinical decision making. The aim of the current debate paper is to further explore the argument that N-of-1 trial designs, and the wider family of single-case experimental designs (SCEDs), offer useful research designs for health practitioners who wish to generate evidence of efficacy of treatment regimens for individual patients. A clinical example of a pragmatic N-of-1 trial protocol is provided and discussed.

### 1.1. Evidence-Based Practice in Context

Evidence-based practice (EBP) is a relatively new construct in Western clinical practice [6]. However, there are historic examples of complementary medicine references to diet in the bible (Book of Daniel circa 150 BC) and for the herb ginseng in the Ben Cao Tu Jing (Chinese Atlas of Materia Medica, circa 1061 AD) [6]. The modern use of the term EBP was first coined in the early 1990s, and by the mid-1990s, Sacket had, now famously, described evidence-based medicine as follows:

It’s about integrating individual clinical expertise and the best external evidence … in making decisions about the care of individual patients [7].

Sacket was ultimately concerned with the care of individuals. He went on to describe RCTs and systematic reviews of RCTs as ‘gold standard’ evidence for treatment interventions, and conceded that in cases where no RCTs had been conducted, lower levels of evidence should be sought. Sacket’s contribution was to provide a systematic approach for practitioners to consult evidence that would inform better practice. Initial criticisms of evidence-based medicine argued that this approach would devalue the importance of clinical expertise and intuition in favor of science and evidence, calling up the age-old rationalism versus empiricism epistemological debate [8,9].

In reality, the uptake of EBP in modern medicine has been slow. It has been estimated that less than half of physicians practice according to evidence-based guidelines even when they are available [10], and less than one-third use electronic databases to find relevant scientific evidence to inform practice [11]. It is unknown how much of the complementary medicine workforce uses evidence to inform their practice. According to a national workforce survey, only 25% of respondents reported using peer-reviewed journals as a source of clinical information. The main sources of clinical information for complementary medicine practitioners were professional association newsletters, magazines or journals (66%), professional association conferences (63%), academic textbooks (52%) and internet sources (52%) [12]. A more recent cross-sectional survey estimated that around 14% of health professionals regularly engaged with EBP across a range of disciplines [13].

Nevertheless, acceptance of the use of evidence to inform clinical practice has gained considerable momentum over the past two decades in Western medicine and is today expected practice from all health professions, including complementary medicine. It has been observed that within the framework of evidence-based medicine:

There is no alternative medicine. There is only scientifically proven, evidence-based medicine supported by solid data or unproven medicine, for which scientific evidence is lacking [14].

From this perspective, a central role for evidence means that it can act as an equalizer in decisions about whether an intervention is safe and effective. Of course, evidence needs to be ethical, but it need not be ideologically- or politically-biased. The scientific method of generating evidence aims to minimize human-induced biases from study designs. Those designs higher up the evidence hierarchy are those most able to minimize sources of bias that are likely to have an impact on the findings.

Incorporating practice-based research principles into complementary medicine practice could help to increase the quality of the evidence-base and perhaps the recognition of complementary medicine practitioners in government policies. If complementary medicine practitioners could demonstrate that they practice scientifically and safely, then critics would need to argue with the evidence, not with the ideology or politics.

### 1.2. Origins of the RCT Research Design

At the time, when innovations in organic chemistry made way for the development of early antibiotics, the major cause of mortality was infectious diseases. With the help of clinical trial methodology, the dominance of pharmaceutical interventions as a panacea for medical conditions first emerged. The first randomized clinical trial took place in 1947. It investigated the use of streptomycin for pulmonary tuberculosis compared to normal care (bed rest) [15]. This trial was remarkable for its meticulous use of allocation concealment and blinded assessors to enable objective testing of a drug therapy for the condition.

The use of a placebo as a comparator condition and other RCT refinements followed, with important developments of ethical guidelines and regulatory frameworks to protect participants and patients from unscrupulous researchers [16]. Rapid progress in clinical trial design development was driven by the pressing need to demonstrate safety and efficacy of new pharmaceutical interventions, which typically countered a specific symptom or disease state.

### 1.3. RCTs and Their Relevance to Clinical Practice

The contemporary challenge to health services in developed nations is the increasing prevalence of chronic diseases. A proportion of those with chronic disease will experience comorbidities. In Australia, 29% of people over 65 years reported having three or more chronic diseases [17]. Concerningly, those with multiple comorbidities experience significantly higher barriers to accessing healthcare [18]. These issues are compounded in ageing and increasingly overweight populations [19]. Such complex chronic conditions are often managed with a number of concomitant pharmaceuticals, known as polypharmacy [20].

A report of community prescribing practices in Tyneside, Scotland in 2010, found that 20% of adults were dispensed more than five pharmaceuticals and almost 7% were dispensed more than 10. Increasing age was the strongest predictor of polypharmacy [21]. Potentially serious adverse events occurred in over 10% of adults and this was strongly associated with the number of drugs dispensed. The report looked at 15 years of prescribing and noted a dramatic increase in rates of polypharmacy since 1995. Observational studies from USA, Canada, Australia and Italy also report high levels of polypharmacy [20,22]. Such high levels of polypharmacy raise concerns about the safety of un-trialed combinations of treatments for individual patients [23,24]. National regulations aim to ensure that the pharmaceuticals prescribed medications have been tested and found to be safe and effective, but this process relies upon RCT methodologies, with a single drug as the intervention in one condition at a time.

Findings from studies with such tightly controlled clinical samples, however, cannot be readily generalized to individuals with complex conditions who necessarily fall outside of such study criteria. Consequently, there has been a shift in interest about the effects of interventions from questions of efficacy (effects in populations estimated from clinical trials) to effectiveness (effects in real world settings). This has led to the search for more ‘pragmatic’ RCT designs. This shift towards more pragmatic evidence coincides with a wider movement towards individualizing health treatments in clinical practice. Innovative experimental methodologies are now required to measure the effectiveness of tailored health care interventions in individuals.

### 1.4. Single Case Experimental Designs (SCEDs)

N-of-1 trials and the wider family of single-case experimental designs (SCEDs) are adaptive and could be well-suited to practitioners who wish to find the optimal treatment interventions for the whole person, including individual patients who may not normally be included in clinical trials. Criticisms of group clinical trial methodology include the tight study criteria that generally exclude patients with multiple co-morbidities, complex conditions and polypharmacy. They also tend to exclude patients who are likely to be non-compliant with the study protocol, such as those diagnosed with serious mental health disorders. Consequently, patients who are non-gender binary, culturally diverse, or marginalized in other ways, tend to be under-represented in clinical trials and systematic review evidence. Most of the early stress research, for example, was conducted on bright young healthy male medical students in samples that provided excellent internal reliability but little applicability to the general population. Group clinical trials are expensive and are designed to estimate population means. As such they are not necessarily representative of most individual patients.

There have been calls for complementary medicine to undertake more N-of-1 trials [25] and to seriously develop an EBP approach [26]. The SCEDs family of methodologies could provide a clear scientific methodological framework that could help the complementary medicine professions, through its researchers and practitioners, to overcome many of the barriers that have prevented evidence generation to support complementary medicine practice, including the complex nature of complementary medicine interventions, the extended time spent with patients, limited complementary medicine research expertise and lack of research funding to conduct large scale clinical trials [27].

The N-of-1 trial design is a subgroup of the RCT methodology. It is an experimental design that involves a single subject being exposed to multiple conditions over time. It resembles a cross-over RCT, but it is just for one person at a time. For instance, a participant may be exposed to an intervention and a control condition, multiple times, incorporating withdrawal (i.e., wash-out) periods, in a randomized sequence, such as an AB BA design [28]. All the experimental protocols of an RCT can be utilized. For instance, there must be concealment of the randomization sequence and, where possible, masking of the interventions from participants and researchers (double-blinding). Follow-up some months after the trial is also recommended. The aim is to compare different treatment regimens to determine the most effective therapeutic intervention for the individual participant. This methodology is particularly useful for patients who are traditionally excluded from group clinical trials, such as patients with rare diseases, comorbidities and polypharmacy regimens [29].

The N-of-1 trial design is gaining in population as modern medicine moves towards patient-centered care, with an emphasis on individualized medicine [30]. According to the Oxford Centre for Evidence-Based Practice, Level 1 evidence is a systematic review of randomized trials or n-of-1 trials; Level 2 evidence is a randomized trial or observational study with dramatic effect [31]. This elevation of the N-of-1 trial design is a remarkable development in the shift towards the clinical importance of personalized patient care over the statistical significance and population effect sizes given by data derived from mean differences between groups.

Following this recognition of N-of-1 trial methodology, a CONSORT statement (Consolidated Standards of Reporting Trials) extension for reporting N-of-1 trials (CENT) was developed [29]. This statement will help practitioners evaluate the quality of such trials and will help to standardize reporting of trials. For instance, these trials are also called N = 1, individual-patient and single-patient trials in the literature. The CENT checklist (Section 1) specifies that the term N-of-1 trial is used in the title. Other important features to note are that there should be more than five measurement points in each period and more than two repeated blocks (see design templates below). Standardization of the quality of reporting will also facilitate the later aggregation of multiple N-of-1 trials into a systematic review of N-of-1 trials. Aggregation of multiple trials into a meta-analysis could facilitate the generalizability of the findings by providing population estimates of effect sizes for the treatment.

Finally, there could be tangible benefits for an N-of-1 trial for patients: (i) they should be involved in the design of the trial, particularly regarding the interventions and outcomes to be assessed, placing them at the center of the research, and (ii) the results can be made in a timely fashion and inform immediate treatment directions. A longer-term follow up ensures that on-going effects of the treatment are objectively monitored, and therefore, it is more likely to capture any sustained or adverse events [32]. For example, patients who are taking very expensive supplements may wish to find out whether they are really working better than a placebo, or a patient may wish to know whether they really need the higher dose.

### 1.5. N-of-1 Trials Versus RCTs for Complementary Medicine

N-of-1 trial designs are much more accessible for complementary medicine practitioners than RCTs. The pharmaceutical model relies on RCTs to show that the pharmacodynamics effect of a drug causes a dose-dependent change either to the cellular metabolism of the patient or to the invading microbe or tumor cell. Drug leads become pharmaceutical medicines only after safety and efficacy are established and the heterogeneity of individual responses to the intervention is accounted for in an RCT design by using sufficient sample sizes. In such scenarios, only specific aspects of the patient’s physiology are predicted to respond in order for the medication to take effect.

This is counter to the way complementary medicine interventions are perceived to work. A single treatment is not generally expected to resolve a patient’s health issues on its own. Complementary medicine practitioners are generally focused on the person, rather than the disease. The holistic approach includes developing a therapeutic relationship between the person and the practitioner, in which the whole person is taken into account. This biopsychosocial approach is now widely adopted by many health professionals. While complementary medicine treatments include treatment for the immediate presenting symptoms, the holistic approach also aims to explore with the person to help them to understand possible underlying causes for the presenting symptoms. Treatment is then tailored to combining several interventions designed to improve the patient’s physiological functions as well as mitigate specific symptoms. Given the differences in the treatment approaches between conventional/reductionist and holistic approaches to treatment interventions, different study designs are needed to detect and measure change within the holistic approach setting. A summary of RCT designs versus N-of-1 trial designs is provided in Table 1.

### 1.6. Governance of N-of-1 Trials

While N-of-1 trials and SCEDs methodology have much to offer, there are many methodological and governance issues that need to be addressed in order to apply these methodologies in clinical practice [33]. Fortunately, many of the issues can be overcome with careful planning of study designs to suit the individual patient holistically and/or complex interventions. For instance, if an intervention is likely to have a curative or long-term sustained effect, such as probiotics for the microbiome, then modifications to the methodology such as the multiple baseline designs (MBD) can be adapted [34]. MBDs are better suited to complementary medicine interventions with a long metabolic half-life and avoid long withdrawal/wash-out periods where the patient may not be receiving any treatment at all [35]. However, not all patient conditions or interventions will be suited to single case experimental design methodologies. These issues can be best thought through in collaborations between complementary medicine practitioners and researchers with SCEDs methodological expertise.

#### 1.6.1. Research Ethics

N-of-1 trials are a useful practice management tool for health practitioners and patients to work together to determine the optimal treatment regimen. Under such circumstances, this trial is considered an extension of usual clinical practice; thus, approval from a Human Research Ethics Committee (HREC) is not required. However, if the purpose of the N-of-1 trial is to publish the study in order to share the findings with other practitioners or to use the results in a secondary analysis, then, this trial is considered to be research. Thus, there is an ethical and legal requirement to seek the prior approval of a HREC. Ultimately, practitioners who wish to produce evidence to support their practice and profession will be required to publish and therefore should seek prior ethical approval before conducting trials in practice. All research involving humans falls under legally–binding ethical principles, as first set out by the Nuremberg Code and the Declaration of Helsinki. Health practitioners seek such approvals from a local university or hospital ethics committee.

#### 1.6.2. Insurance

Professional indemnity insurance provides legal cover for the advice and services provided by health professionals. In Australia, naturopaths are required to hold current professional indemnity insurance by their professional associations, hold a current first aid qualification and meet their continuing professional education requirements.

Those conducting clinical trials in association with universities are covered under the universities’ clinical trials insurance that includes ensuring that sponsors of clinical trials meet the requirements for commercial sponsors, protection for researchers and not-for-profit sponsors. The University’s Clinical Trial Protection covers the University’s legal liability for damages or compensation as a result of any claim or claims made by research subjects in connection with clinical trials approved by HREC and undertaken by the University. All people who are engaged in undertaking the trial are protected, including staff and students.

Protection is likely to be subject to certain terms, exclusions, conditions and limitations. These may affect how and when protection is granted and the amount of any payment. Practitioners intending to undertake an N-of-1 clinical trial for research purposes should consult their HREC to ensure that they have insurance cover for their trial.

#### 1.6.3. Oversight

Good clinical trials governance includes interim monitoring to provide an objective oversight of the safety for patients (e.g., adverse events), the quality of the data being collected and monitoring of treatment effects. This may take the form of a data and safety monitoring board or a treatment effects monitoring committee. The main aim of such monitoring is to make independent decisions about whether and the trial should be stopped early [36].

A clinical trial may be stopped early for a variety of reasons. For example, the treatments may be found to be convincingly different by independent impartial experts. The decision to stop the trial in this case is to prevent the patients in the control group from wasting unnecessary time on an inferior treatment. Other reasons may include that there is early evidence of no convincing difference between groups, unacceptable or toxic side effects of a treatment and poor-quality data, to name a few.

In the event of an N-of-1 trial or series being conducted for research purposes, it is good governance to incorporate an independent committee to monitor the incoming data for patient safety, quality of the data and treatment effects. Clinical trial monitoring committees normally include 3–10 members depending on the size of trial. Small N-of-1 trials should aim for 2–3 members. Membership should be multi-disciplinary and include trial methodologists, ethicists, clinicians and other researchers. For example, in an N-of-1 trial in mental health, an independent clinical psychologist and statistician could provide independent monitoring for the trial.

## 2. Practice-Based Clinical Scenarios in Complementary Medicine

### 2.1. Clinical Scenario: Probiotics for Fibromyalgia

The aim of this project was to assess whether a three-week course of probiotics, as an adjunct with regular treatment, is effective for reducing the pain associated with fibromyalgia compared with a placebo, in a older female patient with fibromyalgia who also experienced bowel symptoms.

#### 2.1.1. Trial Design

This trial incorporated three randomized blocks/pairs, each consisting of 2 × 3-week periods, comprising either the active or placebo supplement (designated A or B). Between each period there is a 2-week washout period. A simplified schemata (i.e., with two blocks) is shown in Figure 1. In N-of-1 trial terminology, a period is the time during which a single treatment (e.g., A or B) is administered. The order of periods within a treatment pair or block is randomized where practicable. A block or pair is a repeated unit of a set number of periods (for example, three repeating blocks/pairs may take the form: AB BA BA). Multiple pairs or blocks (i.e., more than two units) comprise the entire sequence.

#### 2.1.2. Primary Outcome Measurement

The primary outcome was self-reported pain. This is to be recorded at the same time each day during the trial by the patient (preferably that time when pain scores are typically higher). The measure was a 10-point visual analogue for pain that the patient was asked to complete each day at the same time, in response to the question: how much pain are you experiencing right now?

#### 2.1.3. Primary Hypothesis

The primary hypothesis was that the level of pain associated with fibromyalgia will be lower during the active intervention periods compared with the placebo periods.

#### 2.1.4. Statistical Analysis

The primary method of analyzing the data was by systematic visual graphing of the pain scale over the exposure/placebo conditions. This is the method recommended by the CONSORT collaboration, extension for N-of-1 studies (CENT) [37]. This was augmented with a range of secondary outcomes including the patient perception reports on which condition worked best.

#### 2.1.5. Process Evaluation

In this clinical scenario, the treatment intervention was probiotics, which needed a long period of time for the supplement to take effect. With the minimum of three exposures [29], with washouts, it became a very long trial. This was considered a limitation of the trial. The longer the duration of a trial, the more likely that other potentially confounding variables may influence the outcome. The basic AB BA design can be readily modified for shorter acting interventions by reducing the time periods. Alternatively, other single case experimental and non-experimental study designs have been designed to deal with these issues. Most notably a multiple baseline design (MBD). MBDs are indicated where the intervention may cause a sustained effect, causing long washout periods or confounding in the crossover condition. Many allied health interventions teach new skills that cannot be unlearned (e.g., in occupational therapy), or make changes that are not readily or ethically withdrawn. In these cases, a multiple-baseline design (MBD) may be the design of choice to test the effectiveness of an intervention for an individual [38].

## 3. Discussion

N-of-1 trials and SCEDs are readily designed around individual patients, and can take patient perspectives, values and preferences into account. Patient involvement in the design and implementation of the trial helps to give patients a voice to validate their values and preferences, increases patient understanding of their condition, promotes better communication with practitioners and, importantly, helps to provide evidence to guide treatment decisions about outcomes that are meaningful for the patient [39]. By focusing on the individual, rather than group means, N-of-1 trial methodology uniquely accommodates the inherently person-centered nature of a holistic approach to health practice.

### 3.1. Patient-Centred Approach

The idea of placing the patient at the center of their care is not new to complementary medicine practitioners, who have long known this approach as holistic. Patient-centered care has been widely promoted to improve the quality of care. It aims to build a trusting patient-practitioner relationship by valuing the patient’s health and illness experience, their needs and preferences. A systematic review of 34 papers [40] describing patient perceptions of clinical care concluded that complementary medicine consultations provided patients with experiences of empathy, empowerment and patient-centeredness.

A study investigating whether N-of-1 trials can improve patient management concluded that N-of-1 trials were effective particularly when optimal patient management was uncertain [41]. Experimenting with different treatment regimens has the potential to save on the costs of ineffective treatments or treatments that are associated with low patient preference, satisfaction or compliance. For instance, an N-of-1 study embedded into an RCT helped to increase compliance with the study protocol and prevented the withdrawal of a participant [42,43]. Similarly, a systematic review of N-of-1 trials for depression found evidence relevant for treatment-resistant patients and patients with comorbidities who would normally have been excluded from conventional clinical trials [44].

Clough et al. [45] assessed the suitability of N-of-1 trials as method of mitigating the risk of overprescribing by identifying medications that were underperforming for individual patients. They conducted a systematic review and identified six studies published between 1983 and 2005, where the efficacy of an established treatment in older individuals was objectively tested using an N-of-1 design. In four trials, the tested intervention produced non-significant benefits or was poorly tolerated by individual patients and led to discontinuation of the medication, and in two, the treatment was beneficial and patients continued using the medication. One trial lacked a follow-up, significantly limiting the assessment of the decision to de-prescribe and its safety. The authors concluded that the method was appropriate for older patients to discern the effect of de-prescribing medications with short-term outcomes.

A series of three N-of-1 trials in Traditional Chinese Medicine (TCM) illustrates the value of this design in differentiating treatment approaches for individual patients [46]. In the absence of pharmacokinetic evidence about the medicinal herb involved, the authors experimentally determined appropriate onset and washout characteristics prior to the commencement of the trial. Huang et al. enrolled three individuals with stable bronchiectasis in a 3-cycle, 18-week randomized, triple blind N-of-1 study. Patients were exposed to a standard decoction of TCM herbs designed to reduce general symptoms of bronchiectasis such as sputum production in one condition and a decoction of herbs prescribed to each individual symptom picture according to TCM principles in the second condition. All three patients benefitted from the herbal interventions in both conditions; however, one patient experienced relief of concomitant symptoms only on the individualized herbal decoction and sought to stay on this prescription after the trial was completed.

N-of-1 trials have been used to assess therapeutic options in the most vulnerable populations. To assess the effectiveness of a psychostimulant (methylphenidate) on fatigue in patients with advanced cancer, a group of researchers used N-of-1 trial methodology to halve the sample size required to estimate population effects [47]. Forty-three patients were enrolled with 24 completing three random cycles (active and placebo periods). Aggregating individual N-of-1 results from a total of 84 cycles using Bayesian methods provided stable population estimations. The study concluded that the intervention was not effective in the population of advanced cancer patients with fatigue. Interestingly, the trial also allowed for an exploration of individual effects, which is not usually available from RCT trial data. For eight individuals, the intervention did significantly improve fatigue, but for one patient, fatigue was significantly worsened by the intervention. This study illustrates that N-of-1 trials can provide population estimates with small sample sizes, while also allowing important individual responses to be identified and tailoring of appropriate individual treatments.

### 3.2. Limitations of N-of-1 Trials as a Research Methodology

One limitation of the application of N-of-1 trials methodologies for complementary medicine, particularly naturopaths, is that this methodology is best suited to treating the symptoms of chronic, symptomatic diseases. Multiple exposures and withdrawal periods increases rigor due to increasing replications of the effect, culminating in more confidence and more objective assessment of treatment efficacy. Indeed, in N-of-1 trials, the power comes from the number of measurement points, rather than the number of participants.

However, one of the core principles of naturopathic medicine is ‘Treat the Cause (*tolle causam*)’ [2], rather than the symptoms. In theory, this was the case with the clinical scenario described above, using probiotics to correct an imbalance in the microbiome. This is an area of some ideological conflict between the N-of-1 trials design and the philosophical approach by complementary medicine professions. The N-of 1 trial design is best suited for treating symptoms, specifically chronically occurring symptoms, not causes of disease. Thus, N-of-1 trial designs seem to be most indicated in chronic, symptomatic, non-fatal conditions, using treatments that have a relatively short-term duration of action [48].

Fortunately, SCED variants such as the multiple baseline design (MBD) have been developed to accommodate interventions that are designed to cure and are thus not easily ‘withdrawn’ [34]. Examples include interventions with learned behaviors, where a new skill (e.g., posture retraining or a communication strategy) cannot be readily unlearned, or where it may not be practical or ethical to withdraw an intervention once given. This may be particularly relevant to those treatments aimed at remedying the underlying causes of a disease, such as a nutritional intervention to correct an underlying nutritional imbalance. In such cases the multiple baseline design can help to overcome the issue of sustained effects that may carry over into subsequent withdrawal periods (that are meant to be wash-outs or a different intervention). However, the MBD comes at a loss to the high-level experimental design, as the sequence is not fully randomized and cannot be fully blinded to those who are aware of the design. If rigorously applied, therefore, MBD designs could produce quality evidence, but caution should be applied to labelling quasi-experimental designs as level 1 evidence.

A commonly expressed limitation of N-of-1 trial designs is the inability to estimate effect sizes in the population. However, results from multiple N-of-1 trials or a planned series of N-of-1 trials can be combined to estimate population effects [49]. While there is still a lot of methodological work to be done in this area, there is general agreement that Bayesian methods offer an acceptable level of precision in the estimation of parameter estimates [50]. Combining data from multiple N-of-1 trials into secondary meta-analyses to provide generalizable effect sizes could provide a new avenue for a more rapid creation of clinical knowledge than can be gained from long-term clinical trials and subsequent meta-analyses. It remains to be seen whether the estimates are as accurate as those produced in traditional clinical trials, but this is an exciting new source of emerging evidence.

The Hawthorne effect is a possible source of bias that may affect the generalizability of clinical research to clinical practice [51,52]. It is conceivable that this could become an issue in N-of-1 trial designs, particularly meta-analyses of N-of-1 trials and/or in Bayesian approaches. Given the nature of the communication between practitioner and participating patient in the design and implementation phases of an N-of-1 trial, the effect of high-quality attention provided in the therapeutic relationship may itself be an independent effect that should be investigated and controlled for, experimentally or statistically, where possible. For instance, the addition of another condition where the patient receives neither the placebo nor the intervention may allow assessment of this effect.

## 4. Conclusions

As modern medicine moves towards patient-centered models of healthcare, with an emphasis on individualized medicine, the N-of-1 trial design is gaining in popularity and is well suited to test the safety and efficacy of combinations of complementary medicine or integrative treatments [30]. N-of-1 trial designs have the potential to generate the highest quality of evidence about the effectiveness of complementary medicine treatments in individual patients in clinical practice. The challenge may be forming collaborations between researchers and complementary medicine practitioners to ensure adequate training and rigorous trial design and management with independent monitoring.

There are many issues associated with developing an evidence base for complementary medicine. However, the N-of-1 trial design and its many SCED variants may provide a mechanism to overcome many of these issues. This family of research designs are highly suited to holistic clinical practitioners who wish to provide evidence of the effectiveness of their practice with individual patients. The strong evidence base that could arise from widespread production of methodologically-sound clinical research could have a vast impact on the policy and practice of complementary therapies and its contributions towards better health care.

## Figures and Tables

**Figure 1 healthcare-08-00015-f001:**
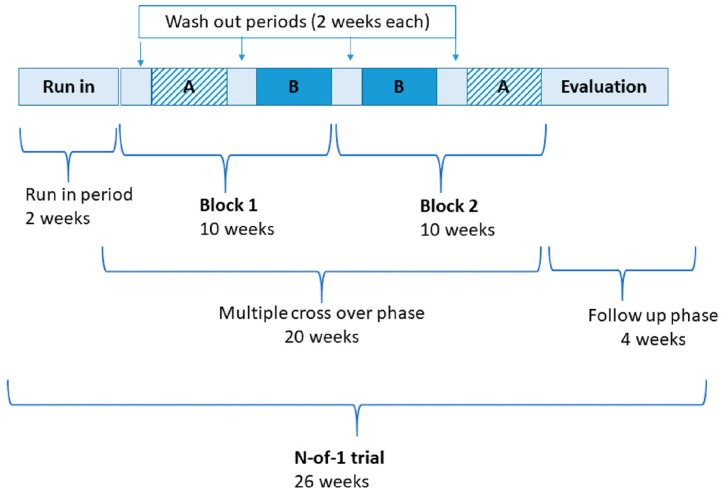
Simplified design of probiotics in fibromyalgia N-of-1 trial with timeline. A—denotes active supplementation; B—denotes placebo supplementation.

**Table 1 healthcare-08-00015-t001:** Comparison of RCTs and N-of-1 trials for clinical effectiveness studies.

	RCT	N-of-1 Trials
**Advantages**	Experimental design to determine cause effect relationship for the intervention on the outcome in carefully selected sample.	Experimental design to determine best intervention for individual patient.
	Tightly controlled clinical environment, increases internal validity.	Patient-centered research, through shared decision making about the study design (e.g., outcomes and/or interventions may be chosen by patient). For example, patients need not withdraw from their usual care, which can be incorporated into the design as a baseline or placebo condition.
	Bias minimized via random allocation to groups, allocation concealment and ongoing blinding of participants and data collectors.	Bias minimized via random allocation of exposure to treatments, allocation concealment and ongoing blinding of participants and data collectors to condition, where possible.
	Effect size estimated and generalizable to populations.	Determination about whether a particular treatment works for an individual at a given point in time.
	Powerful statistical analysis that enable conclusive determinations based on experimental hypothesis testing in adequately powered study designs, based on number of participants.	Power is derived from number of measurement points rather than number of participants.
	Can be included in systematic reviews and meta-analyses of RCTs.	Can be included in systematic reviews and meta-analyses of N-of-1 trials.
	More concerned with efficacy than effectiveness.	More concerned with effectiveness than efficacy.
**Limitations**	Results apply to population means rather than individuals.	Results apply only to the specific individual who was included in the trial.
	Strict inclusion/exclusion criteria means that the sample is not necessarily reflective of clinical usage in a general clinical population (i.e., increasing internal validity reduces the generalizability/external validity of the findings).	Lack of generalizability to estimate effect size in populations in single N-of-1 trials. However, multiple N-of-1 trials can be aggregated as an N-of-1 series or meta-analysis, in order to estimate population effect sizes.
	Expensive and time consuming to run.	Time consuming for practitioner and patient.
	Results often not known for years	Statistical analysis not as powerful as parametric tests are not suited to small number and repeated measures samples (usually violate assumptions of normality and independence); usually uses simple visual descriptive analyses or more complex Bayesian analyses.

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
