# Peer review of "Practice-Based Research in Complementary Medicine: Could N-of-1 Trials Become the New Gold Standard?"

_healthcare, 2020, doi:10.3390/healthcare8010015_

Round 1

Reviewer 1 Report

The changes that were made improved the paper and solved the problems/questions that were identified.

This manuscript is a resubmission of an earlier submission. The following is a list of the peer review reports and author responses from that submission.

Round 1

Reviewer 1 Report

The paper deals with a very relevant and present issue to public health and healthcare. There is a set of questions which I feel need clarifying and should be considered before publication, that support my recommendation that the papers needs major revision.

The authors state that “Rigorously designed and conducted N-of-1 trials could become a new ‘gold standard’ for demonstrating treatment efficacy for complementary medicine interventions in individual patients in clinical practice” (lines 23-25). In the whole article, the authors present arguments in favor of using N-of-1 trials as a tool to demonstrate the effectiveness of complementary medicines. However, I don’t find enough evidence in the paper to support, without doubts, those statements. Since the paper is a review and the authors admit that more research is needed, I would recommend that these arguments could assume the form of questions for discussion, more than a statement pro N-of-1 trials as a solution to the problem of evidence in complementary medicine. Complementary medicine comprises a diverse set of medicines and therapies, with different approaches and different treatment techniques. The paper treats the different therapies/medicines as something uniform, regardless of these differences. Can trials apply equally to homeopathy and naturopathy, for example? A reflection on the specificities of therapies would be important. If the article focuses on naturopathy, as seems to be the case, this focus should be taken early on. A brief explanation of the type of treatments used in naturopathy would also be important. In “1.6.2. Research ethics”, the authors state that “Under such circumstances, this trial is considered an extension of usual clinical practice, thus approval from a Human Research Ethics Committee (HREC) is not required” (lines 278-280). It does not seem relevant to raise this question, if N-of-1 trials are beeing perspectived, in the whole paper, as a tool to demonstrate the effectiveness of complementary medicines or therapies. In that way, N-of-1 trials are considered research. Section 2 “Practice-Bases Clinical Scenarios in Complementary Medicine” would benefit with an introduction paragraph to briefly explain the cases that are going to be presented. It is important to assume them as an illustration of the process of N-of-1 trials and it is also important to situate them in a particular medicine/therapy. The authors point to the long period of the trial as a limitation (line 361). This is something that deserves discussion. As the treatments of complementary medicine have many times results in the medium or long term, how can a trial cope with this limitation? I would like to see this question considered in the “Discussion” section.

Reviewer 2 Report

Authors have discussed a very important issue on research design for complementary medicine in this manuscript. It will assist the dissemination of n=1 trial design to wide population. However, the following concerns for authors to consider and amend the paper:

it seems that authors indicate n=1 trial is gold standard; but later on they add MBD design to n=1 trial. authors also indicate series (p.8/16) which is a completely different study design. I believe authors should focus on n=1 trial only and should not introduce any other design (eg. MBD or case series) into this paper as it is not within the scope of this paper and causes confusion. the title is about "complementary medicine"; however, the examples given are about naturopathy. the title should reflect the contents rather than too broad and cause misleading. It's recommended "Practice-based research in naturopathy: Are n of 1 trials the new gold standard?"  also in the text, it should use naturopathy to replace complementary medicine as complementary medicine consists of about 20 modalities.  please amend some descriptions in the paper:  p.5 last para: "...focused on treating the person, not the disease". This statement is not clear. All therapies treat the person, treat person's conditions, signs and symptoms.  Table 1: need to line up RCT and n of 1 trials columns to make reading easier. 1.6.1 It is not about rationale as it's been explained - to change to scenarios grammars to be checked for 2.1.1, 2.1.2, 2.1.3 (eg. compare to not compare with) errors to be corrected: 2.2. Clinical Scenario 21 should be 2 2.1 and 2.2 "Scenario" change to "Study" Figure 2: all abbreviations to be explained